# Multi-User PIR with Cyclic Wraparound Multi-Access Caches [note 1]

**DOI:** 10.3390/e25081228

**Published:** 2023-08-18

**Authors:** Kanishak Vaidya, Balaji Sundar Rajan

**Affiliations:** Department of Electrical Communication Engineering, IISc Bangalore, Bengaluru 560012, India; kanishakv@iisc.ac.in

**Keywords:** coded caching, private information retrieval, multi-access caches

## Abstract

We consider the problem of multi-access cache-aided multi-user Private Information Retrieval (MACAMuPIR) with cyclic wraparound cache access. In MACAMuPIR, several files are replicated across multiple servers. There are multiple users and multiple cache nodes. When the network is not congested, servers fill these cache nodes with the content of the files. During peak network traffic, each user accesses several cache nodes. Every user wants to retrieve one file from the servers but does not want the servers to know their demands. This paper proposes a private retrieval scheme for MACAMuPIR and characterizes the transmission cost for multi-access systems with cyclic wraparound cache access. We formalize privacy and correctness constraints and analyze transmission costs. The scheme outperforms the previously known dedicated cache setup, offering efficient and private retrieval. Results demonstrate the effectiveness of the multi-access approach. Our research contributes an efficient, privacy-preserving solution for multi-user PIR, advancing secure data retrieval from distributed servers.

## 1. Introduction

The problem of Private Information Retrieval (PIR), initially introduced in Chor et al. (1995) [1], revolves around the confidential retrieval of data from distributed servers. Users aim to retrieve a specific file from a collection of files stored across these servers while keeping the servers unaware of the file’s identity. Sun et al. (2017) [2] present a PIR scheme that minimizes the user’s download cost. Subsequently, the PIR problem has been addressed in various other settings. For instance, in [3], PIR is studied with colluding servers, and in [4], weakly private information retrieval is studied where some information about the user demand is allowed to be known to the servers. In [5], the user is allowed to have files stored as side information.

Currently, PIR is being explored in conjunction with coded caching for content delivery scenarios. As first proposed in [6], the coded caching problem includes a number of users, each with their own cache memories and a single server hosting a number of files. Users fill their caches while the network is not busy, and during periods of high network traffic, they request files from the server. The server broadcasts coded transmissions that benefit multiple users simultaneously. Users can use the content in their caches to decode the files they requested after receiving the broadcasts. A cache-aided PIR technique was recently put up by Caire et al. [7], in which many users, each with access to their own dedicated caches, attempt to privately recover files from non-colluding servers. The advantages of coded caching from [2,6] are combined to provide an order-optimal MuPIR strategy.

In this paper, we use a variation of coded caching known as multi-access coded caching in PIR. In multi-access coded caching, users do not have access to dedicated caches. Instead, there are helper cache nodes, which are accessed by the users. Multiple users can access one helper cache, and a user can access multiple caches. This paper uses a multi-access setup with cyclic wraparound cache access. In cyclic wraparound cache access, the number of users and cache nodes are equal. Multi-access systems with cyclic wraparound cache access are widely studied in the literature. In [8], Hachem et al. derive an order-optimal caching scheme which judiciously shares cache memory among files with different popularities. This idea was extended to a multi-access setup with cyclic-wraparound cache access. In [9], Reddy et al. studied a multi-access coded caching design and proposed a new achievable rate within a multiplicative gap of at most 2 compared to the lower bound for the said problem provided uncoded placement. In [10], a delivery scheme is proposed for the decentralized multi-access coded caching problem where each user is connected to multiple consecutive caches in a cyclic wrap-around fashion. A lower bound on the delivery rate is also obtained for the decentralized multi-access coded caching problem using techniques from index coding. In [11], Cheng et al. propose a transformation approach to generalize the MAN scheme to the multi-access caching systems, such that the results of [8] remain achievable in full generality. In [12], the authors generalize one of the cases in [13], which proposes novel caching and coded delivery schemes maximizing the local caching gain.

**Notation** **1.**
*Consider integers a and b. Then, [a:b]≜{n∈Z|a≤n≤b}. [a]≜[1:a]. For a set S of size |S| and an integer N≤|S|, SN≜{T⊆S:|T|=N}. For set $\left\{a_{n}:n\in \left[N\right]\right\}$ and N⊆[N], $a_{N}$ denotes the set $\left\{a_{n}:n\in N\right\}$. For the set of integers $\left\{a_{i}:i\in \left[N\right]\right\}$ we define ${\langle a_{1},a_{2},\ldots ,a_{N}\rangle }_{C}$ to be the set $\left\{b_{i}:b_{i}=CifC|a_{i},otherwiseb_{i}=a_{i}\;\mathrm{mod}\;C,i\in \left[N\right]\right\}$.*


The following subsections briefly explain the PIR, coded caching and multi-user PIR problems. Firstly, we explain the single-user PIR problem of [2] and introduce the concept of private retrieval from distributed servers. Then, we introduce the coded caching problem [6] and its different variations, i.e., dedicated cache and multi-access coded caching problems. We provide motivation behind the cyclic-wraparound multi-access model in Section 1.3. Then, the combination of the dedicated cache-aided coded caching problem and the PIR problem as described in [7] is introduced in Section 1.4. Finally, the contribution of this paper, which considers a combination of PIR with a multi-access coded caching problem, is summarized in Section 1.5.

### 1.1. Private Information Retrieval

The protocol of Private Information Retrieval [2] allows for the retrieval of a specific file from a set of *N* files $W={\left\{W_{n}\right\}}_{n=1}^{N}$. These files are replicated across *S* non-colluding servers, with each file being of equal size. The objective of PIR is to retrieve the desired file, denoted as $W_{\theta }$, without disclosing its identity to the servers. In other words, the user intends to conceal the index θ from the servers. To achieve this, the user generates *S* queries ${\left\{Q_{s}^{\theta }\right\}}_{s=1}^{S}$ and sends query $Q_{s}^{\theta }$ to server *s*. Upon receiving their respective queries, the servers generate answers based on the query received and the files they possess. Server *s* generates the answer $A_{s}^{\theta }\left(Q_{s}^{\theta },W\right)$ and sends it back to the user. After receiving answers from all *S* servers, the user should be able to decode the desired file. The PIR protocol must satisfy privacy and correctness conditions, which are formally defined as follows:

Privacy condition:$$I\left(\theta ;Q_{s}^{\theta }\right)=0,\;\;\;\;\;\forall s\in \left\{1,\ldots ,S\right\};$$

Correctness condition:$$H(W_{\theta }|\theta ,A_{1}^{\theta }\left(Q_{1}^{\theta },W\right)\ldots A_{S}^{\theta }\left(Q_{S}^{\theta },W\right),Q_{1}^{\theta }\ldots Q_{S}^{\theta })=0.$$

The transmission cost of PIR is defined as
$$R_{PIR}=\frac{\sum_{s=1}^{S}\left(H\left(A_{s}^{\theta }\left(W\right)\right)\right)}{H(W_{\theta })}.$$
A PIR scheme is provided in [2], which incurs the minimum possible transmission cost, $R_{PIR}^{*}$ is given as a function of the number of servers *S* and the number of files is denoted as *N*.
(1)$$R_{PIR}^{*}\left(S,N\right)=1+\frac{1}{S}+\frac{1}{S^{2}}+\ldots +\frac{1}{S^{N-1}}.$$
In the scheme provided in [2], every file has to be divided into $S^{N}$ subfiles, and every server performs a transmission of size $\left(\frac{1}{S}+\frac{1}{S^{2}}+\ldots +\frac{1}{S^{N}}\right)H\left(W_{\theta }\right)$.

**Note**: In the literature, the term rate is used for the transmission cost (e.g., [6]) as we use it here, whereas sometimes (e.g., [2]) the term rate is used for teh mean the inverse of the transmission cost as used by us. We use the term “transmission cost” instead of rate in this paper as in most of the coded caching literature [6,14,15].

### 1.2. Coded Caching

The authors in [6] propose a centralized coded caching system consisting of a server storing *N* independent files $W_{0},\ldots ,W_{N-1}$ of unit size and *K* users with a dedicated cache memory of size *M* files. However, in recent years, multi-access coded caching systems have been gaining attention, where *C* cache nodes exist, and each user can access several of them.

Coded caching systems work in two phases. In the *delivery phase*, which corresponds to low network traffic, the server fills the caches with the contents of the files. Then, in the *delivery phase*, all users wish to retrieve some files from the servers, increasing the network traffic. User *k* wishes to retrieve file $W_{d_{k}}$ where $d_{k}\in \left[0:N-1\right]$. Each user conveys to the server the index of the file they want. The server broadcasts coded transmissions X of size $R_{CC}$ in the unit of files after receiving the user requests. The transmission X is a function of the files stored at the server and user demand. All users should be able to retrieve their chosen files using the caches they can access after receiving the coded transmission X. The quantity $R_{CC}$ is defined as the rate of the coded caching system, and it measures the size of the server’s transmissions.

### 1.3. Multi-Access Coded Caching with Cyclic Wraparound Cache Access

Several approaches exist for users to access cache nodes in multi-access coded caching systems. In [15], multi-access schemes are derived from cross-resolvable designs, and the authors of [14] propose a system where each user can access *L* unique caches, resulting in CL users. This multi-access setting was further generalized in [16], showing that the rate achieved in [14] is optimal for certain cases. In this paper, we focus on a cyclic wraparound cache access approach where C=K and each user accesses *L* neighboring cache nodes. This approach is reminiscent of circular wraparound networks, also known as ring networks, that have been extensively studied in the literature. For example, circular soft-handoff (SH) models in cellular networks [17] arrange nodes (base stations) in a circle, where users access only two nodes, their local node, and the node in the left neighboring cell. Another variant is the circular Wyner model, where nodes are arranged in a circle and users access three nodes (base stations), its local node, and nodes in two neighboring cells. Such settings were studied in [18], and Shannon-theoretic limits for a very simple cellular multiple-access system were obtained. In [19], the Wyner model was studied again, and upper and lower bounds on the per-user multiplexing gain of Wyner’s circular soft-handoff model were presented. In [20], achievable rates were derived for the uplink channel of a cellular network with joint multicell processing. The rates were given in closed form for the classical Wyner model and the soft-handover model. There is extensive research on circular wraparound cache access in multi-access coded caching settings [8,9,10,11,12,13]. Like in cellular networks discussed above, this can occur when cache nodes are arranged in a circular manner and users access the *L* nearest cache nodes.

### 1.4. Dedicated Cache Aided MuPIR

In the dedicated cache setup described in [7], there is a collection of *N* files denoted as ${\left\{W_{n}\right\}}_{n\in [N]}$, which are replicated across *S* servers. The system involves *K* users, each equipped with a dedicated cache capable of storing *M* files. The users aim to retrieve their desired files from the servers. The system operates in two distinct phases.

In the *Placement Phase,* the cache of each user is populated with certain content. This cache content is determined based on the files stored across the servers and is independent of the future demands of the users. Subsequently, in the *Private Delivery Phase,* each user independently selects a file and seeks to privately retrieve their respective file from the servers. To achieve this, the users collaboratively generate *S* queries and transmit them to the servers. Upon receiving their respective queries, the servers respond with answers. The users should be able to decode their desired files using the transmitted answers and the content stored in their caches. In [7], an achievable scheme known as the *product design* is proposed. The product design results in a transmission cost denoted as $R_{PD}$, where
(2)$$\begin{array}{cc} R_{PD}\left(M\right) & =\min\{K-M,R_{PD}^{\prime }\left(M\right)\}\;\mathrm{and} \end{array}$$
(3)$$\begin{array}{cc} R_{PD}^{\prime }\left(M\right) & =\frac{K(1-\frac{M}{N})}{\frac{KM}{N}+1}R_{PIR}^{*}\left(S,N\right) \end{array}$$
whenever M=tN/K for some integer t∈[0:K]. For other memory points, lower convex envelope of points $\left\{\left(M,R_{PD}\left(M\right)\right):M=tN/K,\;t\in \left[0:K\right]\right\}$ is achieved by memory sharing.

Cache-aided multi-user PIR setups with multi-access caches are also considered in [21,22].

### 1.5. Our Contributions

This paper presents a PIR scheme that enables multiple users, aided by multi-access cache nodes, to privately retrieve data from distributed servers. The proposed scheme focuses on the multi-access setup with cyclic wraparound cache access where there are multiple non-colluding servers and all messages are replicated across these servers. The servers are connected with the users through noiseless broadcast links.

The contributions of this paper are as follows.

The paper comprehensively describes the system model for the MACAMuPIR setup with cyclic wraparound cache access. It outlines the key components and mechanisms involved in the scheme.The paper presents an achievable scheme for the multi-access problem described above and characterizes its transmission cost.A comparison is made between the transmission costs of the multi-access setup and a dedicated cache setup proposed in previous work. The results show that the multi-access setup outperforms the dedicated cache setup.The paper includes proofs that validate the privacy guarantees and transmission costs mentioned in the scheme description. These proofs demonstrate the scheme’s ability to preserve user privacy and ensure accurate retrieval of requested data.

### 1.6. Paper Organization

The rest of the paper is organized as follows:In Section 2, the problem statement is described, along with formal descriptions of transmission cost, privacy and correctness conditions.Then, in Section 3, the main results of the paper are summarized. The achieved rate is mentioned in this section.Section 4 has the scheme to achieve the transmission load mentioned in Section 3. We first explain the scheme using a concrete example in Section 4.1. Then, we extend the description to encompass general parameters in Section 4.2. We then specialize the scheme to the context of cyclic wraparound cache access in Section 4.3. Then, proof of privacy and calculation of subpacketization level follows.After the specialized description of Section 4.3, we arrive at the critical observation that to calculate the rate, it is essential to characterize the number of t+L-sized subsets of [K] that contain at least *L* consecutive integers, with wrapping around *K* allowed. Here, t,K,L∈Z. This is calculated in Section 4.4 onward.Section 5 contains a discussion about the results and scope for future research, and Section 6 concludes the paper.

## 2. System Model: MACAMuPIR with Cyclic Wraparound Caches

The system consists of *K* users and *N* independent files, denoted as ${\left\{W_{n}\right\}}_{n\in [N]}$, which are replicated across S≥2 servers. Each file has a unit size. There are *C* cache nodes available, and each cache can store up to *M* files. Each user is connected to a unique set of L≤C cache nodes through links with infinite capacity. User *k* is connected to cache nodes indexed by Lk∈[C]L. The system follows a multi-access setup with cyclic wraparound cache access. In this setup, the number of users equals the number of cache nodes, i.e., C=K. Each user accesses *L* consecutive caches in a cyclic wraparound fashion. Specifically, user k∈[K] accesses caches indexed by $L_{k}={\langle k+l:l\in \left[0:L-1\right]\rangle }_{K}$. We consider Figure 1; it is a multi-access system with cyclic wraparound access. In this system, we have S=2 servers and K=4 users. There are four cache nodes, and every user is accessing L=2 cache nodes. The system operates in two phases described below.

*Placement Phase*: During this phase, all *C* cache nodes are populated. We let $Z_{c}$ denote the content stored in cache c∈[C]. The content $Z_{c}$ is determined based on the files $W_{\left[1:N\right]}$, and all servers possess knowledge of the content stored in each helper cache.

*Private Delivery Phase*: In this phase, each user aims to retrieve a specific file from the servers. User *k* selects $d_{k}\in \left[N\right]$ and desires to privately retrieve $W_{d_{k}}$ from the servers. The demand vector is denoted as $d=(d_{1},d_{2},\ldots ,d_{K})$. To retrieve their desired files from the servers while preserving privacy, users cooperatively generate *S* queries $Q_{s}^{d},s\in \left[S\right]$ based on their demands and the content stored in helper caches. These queries are designed in a way that they do not disclose the demand vector d to any of the servers. After generating the queries, each query $Q_{s}^{d}$ is sent to server s,∀s∈[S]. Upon receiving their respective queries, each server s,∀s∈[S], broadcasts the answer $A_{s}^{d}$ to the users. The answer is a function of the query $Q_{s}^{d}$ and the files $W_{\left[1:N\right]}$. After receiving all *S* answers $A_{\left[S\right]}^{d}$, each user *k*, ∀k∈[K] decodes $W_{d_{k}}$ using the caches accessible to that user.

To ensure the privacy of the user demands, the following condition must be satisfied:$$I\left(d;Q_{s}^{d},Z_{\left[1:C\right]}\right)=0,\forall s\in \left[S\right],$$
This condition, known as the privacy condition, ensures that none of the servers have any information about user demands. And
$$H\left(W_{d_{k}}|d,Z_{L_{k}},A_{\left[S\right]}^{d}\right)=0,\forall k\in \left[K\right],$$
known as the correctness condition, ensures that users experience no ambiguity concerning their desired file.

We define the transmission cost *R* as the amount of data that has to be transmitted by all the servers in order to satisfy the user demand.
$$R=\sum_{s=1}^{S}H\left(A_{s}^{d}\right).$$ Our goal is to design cache placement and private delivery schemes that satisfy privacy and correctness conditions and minimize the transmission cost.

## 3. Main Results: Achievable Rate and Comparison

In this section, we present the main result of the paper. For a given multi-access cache-aided MuPIR problem, we provide a scheme in Section 4 that can privately retrieve files from *S* non-colluding servers. For the multi-access cache aided MuPIR problem with cyclic wraparound cache access, the scheme incurs a transmission cost as described in Theorem 1. Then we compare the results of Theorem 1 with the dedicated cache-aided system of [7].

### 3.1. Achievable Rate

Before stating the transmission cost for cyclic wraparound cache access setup, we define the quantity cyc(n,k,m) for integers m≤k<n as the number of *k*-sized subsets of *n* distinguishable elements arranged in a circle, such that there is at least one set of *m* consecutive elements amongst those *k* elements. An expression for cyc(n,k,m) is given in Equation (Equation 4), the proof of which is given in Section 4.4.
(4)cyc(n,k,m)=∑r=1kn−kr+n−k−1r−1∑l=1r(−1)l−1rlk−l(m−1)−1r−1 +∑r=3kn−k−1r−2(∑l=2m−1(l−1)∑j=1r−2(−1)j−1r−2jk−l−j(m−1)−1r−3 +∑l=mk(l−1)k−l−1r−3)+k−1.

**Theorem** **1.**
*For the cyclic wraparound multi-access coded caching setup, with S servers, N files, K helper caches and K users, where each user is accessing the L helper cache in a cyclic wraparound manner and each cache can store M files and $t=\frac{KM}{N}$ is an integer, the users can retrieve their required file privately, i.e., without revealing their demand to any of the servers, with*

(5)
$$\begin{array}{cc} R(t) & =\min\{R_{PD}\left(M\right),R^{\prime }\left(t\right)\},\;where \end{array}$$


(6)
R′(t)=cyc(K,t+L,L)KtRPIR*(S,N).



**Proof.** In Section 4, we present a scheme that achieves $R^{\prime }\left(t\right)$ as stated above for cyclic wraparound cache access setup. As users in the multi-access setup with cyclic wraparound cache access are accessing the caches, which the users of the dedicated cache setup are also accessing, the transmission cost of the multi-access setup are no higher than that of dedicated cache setup. For instance, if for some *M*, $R_{PD}\left(M\right)<R^{\prime }\left(M\right)$, then the placement and delivery strategy of the product design can be employed. □

Theorem 1 characterizes a transmission cost in a multi-access setup where cache memory *M* is the integer multiple of N/K. For intermediate memory points, lower convex envelope of points
$${\left\{(t,R(t))\right\}}_{t\in [0:K]}$$
can be achieved by memory sharing.

### 3.2. Comparison with the Dedicated Cache Setup of [7]

We conduct a comparison between our scheme for cyclic wraparound multi-access systems and the product design proposed in [7]. In this comparison, we assume that the cache sizes and the number of users are identical in both settings. It is worth noting that the parameter $t=\frac{KM}{N}$ represents the number of times the entire set of *N* files can be replicated across the caches. For example, if t=2, it implies that the cache nodes can store 2N units of data. Additionally, the total memory capacity of the system is tN units, which is equal to KM. It is important to mention that the transmission cost incurred by the product design is the same as the transmission cost presented in Theorem 1 for the special case where L=1, indicating that each user only accesses one cache node.

To compare the transmission costs of both settings, we consider K=8 users, S=2 servers, and N=3 files, and plot the transmission cost for various values of t∈[8] and L∈[7]. The results are depicted in Figure 2. It can be observed that due to the access to a larger cache memory, the multi-access system outperforms the dedicated cache setup in terms of transmission cost.

## 4. Achievable Scheme: Proof of Theorem 1

### 4.1. Example

In this section, we present an achievable scheme using an example. Let us consider a cache-aided system with N=8 files denoted as $W_{1},W_{2},\ldots ,W_{8}$, C=8 cache nodes, and K=8 users. Each user has access to L=3 cache nodes in a cyclic wraparound manner. Since each user is connected to a unique set of three cache nodes, we can index each user with a subset of [8] of size 3. For instance, the user connected to cache nodes indexed by six, seven, and eight can be denoted as {6,7,8}. Here is the list of all eight users:{1,2,3},{2,3,4},{3,4,5},{4,5,6},{5,6,7},{6,7,8},{1,7,8},{1,2,8}.

**Placement Phase**: We let $t=\frac{CM}{N}=1$. We divide each file into 81=8 subfiles, each indexed by integers in [8].
$$W_{n}=\left\{W_{n,1},W_{n,2},W_{n,3},W_{n,4},W_{n,5},W_{n,6},W_{n,7},W_{n,8}\right\}.$$

Then, we fill the cache nodes as follows:$$\begin{array}{cccccccc} Z_{1} & =W_{[8],1\;\;\;\;\;\;} & Z_{2} & =W_{[8],2\;\;\;\;\;\;} & Z_{3} & =W_{[8],3\;\;\;\;\;\;} & Z_{4} & =W_{[8],4}\\ Z_{5} & =W_{[8],5\;\;\;\;\;\;} & Z_{6} & =W_{[8],6\;\;\;\;\;\;} & Z_{7} & =W_{[8],7\;\;\;\;\;\;} & Z_{8} & =W_{[8],8}. \end{array}$$

**Delivery Phase**: In this phase, every user chooses one of the file indexes. We enumerate the demands of the users:$$\begin{array}{cccccccc} d_{\left\{1,2,3\right\}} & =1\;\;\;\;\;\; & d_{\left\{2,3,4\right\}} & =5\;\;\;\;\;\; & d_{\left\{3,4,5\right\}} & =7\;\;\;\;\;\; & d_{\left\{4,5,6\right\}} & =3\\ d_{\left\{5,6,7\right\}} & =1\;\;\;\;\;\; & d_{\left\{6,7,8\right\}} & =2\;\;\;\;\;\; & d_{\left\{1,7,8\right\}} & =8\;\;\;\;\;\; & d_{\left\{1,2,8\right\}} & =4. \end{array}$$

For privately retrieving the files, users cooperatively generate queries as follows. For every S∈[8]4, such that S is the superset of at least one user index set, users generate sub-queries. For example, for S={3,4,5,6}, we have users {3,4,5} and {4,5,6} as a subset of {3,4,5,6}. Therefore, the users generate
$$Q_{s}^{d,\{3,4,5,6\}}≜\left\{Q_{\left[2\right]}^{d_{\left\{3,4,5\right\}},\left\{3,4,5,6\right\}},Q_{\left[2\right]}^{d_{\left\{4,5,6\right\}},\left\{3,4,5,6\right\}}\right\}$$
corresponding to {3,4,5,6}, where $Q_{\left[2\right]}^{d_{\left\{3,4,5\right\}},\left\{3,4,5,6\right\}}$ are the queries generated by the users in a single-user PIR setup if the demand is $d_{\left\{3,4,5\right\}}=3$ and the set of files are $W_{[8],6}$, whereas there is no user for which {1,4,6,8} is a superset; therefore, no queries can be generated corresponding to S={1,4,6,8}.

Then, for $Q_{s}^{d,\{3,4,5,6\}}$, server *s* transmits
(7)$$A_{s}^{d_{4,5,6}}\left(Q_{s}^{d_{4,5,6},\left\{3,4,5,6\right\}},W_{[8],3}\right)+A_{s}^{d_{3,4,5}}\left(Q_{s}^{d_{3,4,5},\left\{3,4,5,6\right\}},W_{[8],6}\right),$$
where $A_{s}^{d_{4,5,6}}\left(Q_{s}^{d_{4,5,6},\left\{3,4,5,6\right\}},W_{[8],3}\right)$ is the answer of server *s* in a single-user PIR setup if the received query is $Q_{s}^{d_{\left\{4,5,6\right\}},\left\{3,4,5,6\right\}}$ and the set of files is $W_{[8],3}$.

#### Decoding

We consider user {4,5,6} and subfiles $W_{3,3}$ and $W_{3,4}$. Subfile $W_{3,4}$ is available to the user from the cache node 4. Subfile $W_{3,3}$ has to be decoded from the transmissions. Consider the transmissions corresponding to S={3,4,5,6} from (Equation 7):

User {4,5,6} has access to subfiles $\left\{W_{[8],6}\right\}$, and therefore it can reconstruct
$$A_{s}^{d_{3,4,5}}\left(Q_{s}^{d_{3,4,5},\left\{3,4,5,6\right\}},W_{[8],6}\right)$$
using the contents in Cache 6. After removing $A_{s}^{d_{3,4,5}}\left(Q_{s}^{d_{3,4,5},\left\{3,4,5,6\right\}},W_{[8],6}\right)$ from the transmission corresponding to {3,4,5,6}, user {4,5,6} obtains $A_{s}^{d_{4,5,6}}\left(Q_{s}^{d_{4,5,6},\left\{3,4,5,6\right\}},W_{[8],3}\right),$∀s∈{1,2}. As these are the answers of a single-user PIR setup for demand $d_{\left\{4,5,6\right\}}=3$ and files $W_{[8],3}$, user {4,5,6} can decode $W_{3,3}$.

### 4.2. General Scheme: *K* Users, Each Connected to a Unique Arbitrary Set of *L* Caches

Consider *N* independent unit size files ${\left\{W_{n}\right\}}_{n\in [N]}$ replicated across the *S* servers. There are *C* cache nodes, each capable of storing *M* files, and *K* users each connected to a unique set of *L* cache nodes. As each user is connected to a unique set of the *L* cache, we index them with an *L*-sized subset of [C]. Specifically, user K, where K∈[C]L, is the user connected to cache nodes indexed by K. We let U be the set of all users where
U∈[C]LK. Note that, for the special case of cyclic wraparound cache access, C=K and $U=\{L_{k}:k\in \left[K\right]\}$ where $L_{k}={\langle k+l:l\in \left[0:L-1\right]\rangle }_{K},\;\forall k\in \left[K\right]$.

**Placement Phase**: We let $t=\frac{CM}{N}$ be an integer. Then, we divide each file into Ct subfiles, each indexed by a *t*-sized subset of [C].
Wn=Wn,T|T∈[C]t.

Then, we fill cache node *c* with
Zc=Wn,T|c∈T,T∈[C]t.

**Delivery Phase**: In this phase, every user chooses one of the file indexes. We let user K,∀K∈U choose index $d_{K}\in \left[N\right]$. User K then wishes to retrieve file $W_{d_{K}}$ from the servers without reveling the index of the demanded file to the servers. We let $d={\left(d_{K}\right)}_{K\in U}$ be the demand vector. Users do not want the servers to obtain any information about the demand vector. For privately retrieving the files, the users cooperatively generate *S* queries $Q_{s}^{d}$ as follows. For every S∈[C]t+L, such that S⊃K for at least one K∈U, the users generate sub-queries
(8)Qsd,S=QsdK,S|K∈SL∩U
where the sub-sub-query $Q_{s}^{d_{K},S}$ is the query sent to server *s* in a single-user PIR setup of [2] if the user demand is $d_{K}$. We note that ${\left\{Q_{s}^{d_{K},S}\right\}}_{s\in [S]}$ for all K and for all S are generated independently. The query sent to server *s* is
(9)Qsd=Qsd,S|S∈[C]t+L,S⊃K for some K∈U.

Now, for every $Q_{s}^{d,S}$, server *s* transmits
(10)⨁K∈SL∩UAsdK(QsdK,S,W[N],S∖K),
where $A_{s}^{d_{K}}\left(Q_{s}^{d_{K},S},W_{[N],S\setminus K}\right)$ is the answer of server *s* in a single-user PIR setup if the received query is $Q_{s}^{d_{K},S}$ and the set of files is $\left\{W_{[N],S\setminus K}\right\}$.

Now, we proceed to show that all the users are able to decode their required file from these transmissions and the caches they have access to.

#### 4.2.1. Decoding

We consider user K (i.e., the user connected to cache nodes indexed by K) and subfile index T. If K∩T≠ϕ, then the subfile $W_{d_{K},T}$ is available to the user from the cache. If K∩T=ϕ, then the subfiles have to be decoded from the transmissions. We consider transmissions corresponding to S=K∪T.
(11) ⨁K′∈K∪TL∩UAsdK′(QsdK,K∪T,W[N],(K∪T)∖K′) =AsdK(QsdK,K∪T,W[N],T)⊕ ⨁K′∈K∪TL∩U∖KAsdK′(QsdK′,K∪T,W[N],(K∪T)∖K′). User K has access to all the subfiles in the second term of RHS above, and therefore it can recover the first term from the above expression. After obtaining $A_{s}^{d_{K}}\left(Q_{s}^{d_{K},K\cup T},W_{[N],T}\right)$ for all s∈[S], user K can recover subfile $W_{d_{K},T}$ from the transmissions.

#### 4.2.2. Proof of Privacy

Now, we show that query $Q_{s}^{d}$ sent to server *s* is independent of the demand vector d, ∀s∈[S]. For some S, we consider $Q_{s}^{d,S}$ in (Equation 8). We show that $Q_{s}^{d,S}$ is independent of the demand vector. From the privacy of the single-user PIR scheme, the demand of user K∈SL∩U, i.e., $d_{K}$ is independent of sub-sub-query $Q_{s}^{d_{K},S}$. Also, other sub-sub-queries in $Q_{s}^{d,S}$ are similarly independent of $d_{K}$ as they correspond to the users other than K. This means that $Q_{s}^{d,S}$ is independent of the demands of the users in SL∩U. Moderover, all $Q_{s}^{d_{K},S}$ for any K and S are constructed independently, so these are also independent of the demands of users in U∖SL. This shows that $Q_{s}^{d,S}$ is independent of the demand vector d. Same analysis is true for any S∈[C]t+L∩U where S⊃K for at least one K∈U, which completes the proof of privacy for our scheme.

#### 4.2.3. Subpacketization

As we can see, each file is divided into Ct subfiles. According to the single-user PIR scheme, each of these subfiles has to be further divided into $S^{N}$ sub-subfiles. Therefore, the subpacketization level is Ct×SN.

### 4.3. General Scheme: Cyclic Wraparound Cache Access

For cyclic wraparound cache access systems, we have C=K and $U=\{L_{k}:k\in \left[K\right]\}$. Therefore, transmissions are performed only for those S∈[K]t+L for which $L_{k}\subset S$ for at least one k∈[K]. This is the same as the number of t+L-sized subsets of [K] that contain at least *L* consecutive integers, with wrapping around *K* allowed. As shown in Section 4.4, there are cyc(K,t+L,L) such subsets of [K]. Now, cyc(K,t+L,L) transmissions are performed by each of the *S* servers, and every transmission is of size 1S+…+1SN/Kt units; therefore, the transmission cost incurred is
R(t)=cyc(K,t+L,L)Kt1+1S+…+1SN−1.

Also, note that user *k* of the dedicated cache setup and that of the multi-access setup with a cyclic wraparound cache access are accessing cache node *k*. In a dedicated cache setup Kt+1, transmissions are required to satisfy user demands. Therefore, when cyc(K,t+L,L)>Kt+1, we perform placement and transmissions as conducted for a dedicated cache setup. In this scenario, the transmission cost incurred in a multi-access setup is only as high as the transmission cost of a dedicated cache scenario with same cache sizes. For t∈[0:K], the transmission cost of a multi-access setup would be
minRPD(tNK),cyc(K,t+L,L)KtRPIR*(S,N).

We demonstrate our claim using an example for K=8 caches and users, S=2 servers, N=3 files and L=2. In Figure 3, we see that for smaller values of the *t* cyclic wraparound, cache access is incurring more transmission cost than the dedicated cache setup. For instance, when M=0.75 or t=2, cyc(8,4,2)=68, transmissions are performed for cyclic wraparound cache access without memory sharing (and incurring transmission cost 4.25) compared to 83=56 transmissions in a dedicated cache setup (and incurring transmission cost 3.5). Therefore, when t=2, transmissions corresponding to dedicated cache setup are performed. But when M=1.125 or t=3, a multi-access system with a cyclic wraparound cache access satisfy user demand with 56 transmissions (and an incurring transmission cost 1.75) compared to a dedicated cache setup which requires 70 transmissions (and sn incurring transmission cost 2.1875), and therefore transmissions, as described here, are performed. For cache memory *M*, 0.75<M<1.125, memory sharing between these two schemes can incur transmission cost lower than either of these schemes.

### 4.4. Proving the Expression for cyc(n,k,m)

In this section, we show that the number of ways of choosing *k* integers from the set [n], such that there is a subset of at least *m* consecutive integers, with cyclic wrapping around *n* allowed, is cyc(n,k,m) as defined in (Equation 4).

First, for every K∈[n]k, we denote $i_{l}$ as the length of $l^{th}$ consecutive runs of integers inside K and $o_{l}$ is the length of $l^{th}$ consecutive run of integers outside K. For instance, if n=10 and K={1,2,4,9,10}, then $i_{1}=2$ corresponding to elements {1,2} in K, $o_{1}=1$ corresponding to {3} not in K, $i_{2}=1$ corresponding to element {4} in K, $o_{2}=4$ corresponding to {5,6,7,8} not in K and $i_{3}=2$ corresponding to {9,10} in K. Now, every K∈[n]k can be uniquely determined by a sequence of positive integers consisting of $i_{l}$ and $o_{l}$ where every integer provides the length of consecutive runs of integers inside or outside K, provided it is known if 1 is inside or outside K. For example, with n=10 and k=6, if we are given the sequence of lengths of consecutive runs of the integers inside and outside K as 3,2,3,2 and it is known that 1∈K, then we can uniquely figure out K={1,2,3,6,7,8}.

Now, the set of all *k*-sized subsets K of [n] with at least *m* cyclically consecutive integers can be partitioned into four disjoint sets as follows:1∈K and n∉K. This corresponds to sequences of the form $i_{1},o_{1},\ldots ,i_{r},o_{r}$ where $i_{l},o_{l}\geq 1$ for all l∈[r], $\sum_{l\in [r]}i_{l}=k$, $\sum_{l\in [r]}o_{l}=n-k$, ∃l∈[r] such that $i_{l}\geq m$, ∀r∈[k−m+1]. We let the set of all such *k*-sized subsets be $K_{1}$.1∉K and n∈K. This corresponds to sequences of the form $o_{1},i_{1},\ldots ,o_{r},i_{r}$ where $i_{l},o_{l}\geq 1$ for all l∈[r], $\sum_{l\in [r]}i_{l}=k$, $\sum_{l\in [r]}o_{l}=n-k$, ∃l∈[r] such that $i_{l}\geq m$, ∀r∈[k−m+1]. We let the set of all such *k*-sized subsets be $K_{2}$.1∉K and n∉K. This corresponds to sequences of the form $o_{1},i_{1},\ldots ,o_{r},i_{r},o_{r+1}$ where $i_{l},o_{l}\geq 1$ for all l∈[r+1], $\sum_{l\in [r]}i_{l}=k$, $\sum_{l\in [r+1]}o_{l}=n-k$ and ∃l∈[r] such that $i_{l}\geq m$, ∀r∈[k−m+1]. The set of all such *k*-sized subsets is denoted by $K_{3}$.1∈K and n∈K. This corresponds to sequences of the form $i_{1},o_{1},\ldots ,o_{r-1},i_{r}$ where $i_{l},o_{l}\geq 1$ for all l∈[r], $\sum_{l\in [r]}i_{l}=k$, $\sum_{l\in [r]}o_{l}=n-k$ and ∃l∈[2:r−1] such that $i_{l}\geq m$ or $x_{1}+x_{r}\geq m$, ∀r∈[k−m+1]. We let the set of all such *k*-sized subsets be denoted by $K_{4}$.

Now, we have $\mathrm{cyc}\left(n,k,m\right)=|K_{1}\left|+\right|K_{2}\left|+\right|K_{3}\left|+\right|K_{4}|$. We proceed to calculate the size of these sets individually.

#### 4.4.1. Calculation of $\left|K_{1}\right|$

Sets in $K_{1}$ correspond to the positive integer sequences of the form $i_{1},o_{1},\ldots ,i_{r},o_{r}$. Here, $\sum_{l\in [r]}i_{l}=k$ and $\sum_{l\in [r]}o_{l}=n-k$, and at least one $i_{l}\geq m$ and *r* takes all possible values in [k−m+1].

We let $I_{j}^{r}$ denote the set of tuples of *r* positive integers with the sum of integers equal to *k* and teh $j^{th}$ integer greater than or equal to *m*, i.e.,
$$I_{j}^{r}=\left\{\left(i_{1},i_{2},\ldots ,i_{r}\right):\sum_{l\in [r]}i_{l}=k,i_{l}\geq 1,\forall l\in \left[r\right],i_{j}\geq m\right\}.$$ For a given *r*, $\cup_{j\in [r]}I_{j}^{r}$ is the set of all *r* length sequences, $\left(i_{1},\ldots ,i_{r}\right)$, of positive integers such that $\sum_{l\in [r]}i_{l}=k$. For all such sequences $i_{1},\ldots ,i_{l}$ there also exist n−k−1r−1 sequences of positive integers $o_{1},\ldots ,o_{r}$ such that $\sum_{l\in [r]}o_{l}=n-k$. Therefore,
|K1|=∑r∈[k−m+1]n−k−1r−1⋃j∈[r]Ijr.
From the inclusion–exclusion principle, we know that
$$|\underset{j\in [r]}{⋃}I_{j}^{r}|=\sum_{l=1}^{r}{\left(-1\right)}^{l-1}\sum_{1\leq j_{1}<\cdots <j_{l}\leq r}\left|I_{j_{1}}\cap \cdots \cap I_{j_{l}}\right|,$$
where
Ij1∩⋯∩Ijl=|{(i1,…,ir):∑l∈[r]il=k,il≥1,∀l,ij1,…,ijl≥m}| =|{(i1,…,ir):∑l∈[r]il=k−l(m−1),il≥1,∀l∈[r]}| =k−l(m−1)−1r−1,
which implies
|K1|=∑r∈[k−m+1]n−k−1r−1⋃j∈[r]Ijr =∑r∈[k−m+1]n−k−1r−1×∑l∈[r](−1)l−1∑1≤j1<⋯<jl≤rk−l(m−1)−1r−1 =∑r∈[k−m+1]n−k−1r−1×∑l∈[r](−1)l−1rlk−l(m−1)−1r−1.

#### 4.4.2. Calculation of $\left|K_{2}\right|$

By the definition of the set $K_{2}$ and from the sequence of integers $o_{1},i_{1}\ldots o_{r},i_{r}$ corresponding to $K_{2}$, it is clear that
$$|K_{2}\left|=\right|K_{1}|.$$

#### 4.4.3. Calculation of $\left|K_{3}\right|$

Here, we again see that we need a sequence of positive integers $i_{1},\ldots ,i_{r}$ such that $\sum_{l\in [r]}=k$ and ∃l∈[r] for which $i_{l}\geq m$. We have already calculated this quantity for $|K_{1}|$, but for every such sequence of integers, there exist n−k−1r sequences $o_{1},\ldots ,o_{r+1}$ of positive integers such that $\sum_{l\in [r+1]}o_{l}=n-k$. Therefore,
|K3|=∑r=1k−m+1n−k−1r∑l∈[r](−1)l−1rlk−l(m−1)−1r−1.

#### 4.4.4. Calculation of $\left|K_{4}\right|$

We consider all sequences of integers $i_{1},o_{1},\ldots ,o_{r-1},i_{r}$ corresponding to $K_{4}$ such that $i_{l},o_{l}\geq 1$ for all l∈[r], $\sum_{l\in [r]}i_{l}=k$, $\sum_{l\in [r]}o_{l}=n-k$ and r≥2 and ∃l∈[r] such that $i_{l}\geq m$ OR $i_{1}+i_{r}\geq m$. $K_{4}$ can be partitioned into two disjoint subsets, $K_{41}$ corresponding to sequences where $i_{1}+i_{r}<m$ and $i_{l}\geq m$ for at lest one l∈[2:r−1] and $K_{42}$ corresponding to sequences where $i_{1}+i_{r}\geq m$. Again, $K_{4}=K_{41}\cup K_{42}$ and $K_{41}\cap K_{42}=\phi$. We proceed to calculate the cardinality of both these sets separately.

#### Calculation of $\left|K_{41}\right|$

We consider the set of all *r* length positive integer sequences $i_{1}\ldots i_{r}$ such that $i_{1}+i_{r}<m$ and $\sum_{l\in [r]}i_{l}=k$ and $i_{l}\geq m$ for some l∈[2:r−1]. We note that, for such sequences, r>3. The number of such sequences is
 {(i1,…,ir):∑l∈[r]il=k,il≥1,i1+ir<m,∃ls.t.il≥m} =∑s=2m−1(s−1){(i2…ir−1):∑l=2r−1il=k−s,il≥1,∃ls.t.il≥m} =∑s=2m−1(s−1)∑j=1r−2(−1)j−1r−2jk−s−j(m−1)−1r−3. For every such *r* length sequence, there exist n−k−1r−2 positive integer sequences $o_{1}\ldots o_{r-1}$ such that $\sum_{l\in [r-1]}o_{l}=n-k$, and we obtain
|K41|=∑r=3k−m+1n−k−1r−2∑s=2m−1(s−1)×∑j=1r−2(−1)j−1r−2jk−s−j(m−1)−1r−3.

#### Calculation of $\left|K_{42}\right|$

We consider the set of all r>2 length positive integer sequences $i_{1}\ldots i_{r}$ such that $i_{1}+i_{r}\geq m$ and $\sum_{l\in [r]}i_{l}=k$. The number of such sequences is
 {(i1…ir):∑l∈[r]il=k,i1+ir≥m,il≥1}=∑s=mk−(r−2)(s−1){(i2…ir−1):∑l∈[2:r−1]il=k−s,il≥1}=∑s=mk−(r−2)(s−1)k−s−1r−3. For every such *r* length sequence, there exist n−k−1r−2 positive integer sequences $o_{1}\ldots o_{r-1}$ such that $\sum_{l\in [r-1]}=n-k$, and when r=2, there are k−1 possible pairs of positive integers $i_{2},i_{2}$ which provides $i_{1}+i_{2}=k$, leading to
|K42|=∑r=2k−m+1n−k−1r−2∑s=mk−(r−2)(s−1)k−s−1r−3+k−1. Finally, we have
 cyc(n,k,m)=|K1|+|K2|+|K3|+|K41|+|K42| =∑r=1k−m+1n−k−1r−1∑l∈[r](−1)l−1rlk−l(m−1)−1r−1 +∑r=1k−m+1n−k−1r−1∑l∈[r](−1)l−1rlk−l(m−1)−1r−1 +∑r=1k−m+1n−k−1r∑l∈[r](−1)l−1rlk−l(m−1)−1r−1 +∑r=3k−m+1n−k−1r−2∑s=2m−1((s−1) ×∑j∈[r−2](−1)j−1r−2jk−s−j(m−1)−1r−3) +∑r=3k−m+1n−k−1r−2∑s=mk−(r−2)(s−1)k−s−1r−3+k−1. Defining ab=0 if a<b or if a<0 or b<0, the expression above can be simplified to (Equation 4).

## 5. Discussion

In this paper, we proposed an efficient and privacy-preserving scheme for multi-user retrieval scenarios. By leveraging the benefits of multi-access setups with cyclic wraparound cache access, we demonstrated improved transmission costs compared to the dedicated cache setup. We conducted a comprehensive comparison with prior works that utilize dedicated cache systems. Our results demonstrate the superior performance of our proposed scheme.

Moreover, the placement and delivery schemes designed for dedicated cache-aided MuPIR scenarios are applicable in our MAC-MuPIR scenarios with cyclic wraparound cache access. This adaptability leads to consistently lower rates in our scheme compared to the product design. Hence, we achieved even lower rates by utilizing memory sharing between our setup and the dedicated cache-aided setup. For instance, if we consider Figure 3, specifically for points t=1,2.5 and 4. At t=1, the dedicated cache setup of [7] achieves a lower rate than the scheme provided in Section 4.3, so placement and delivery, as described in [7], are performed, which still work in our setting. At t=4, the scheme provided in Section 4.3 has a lower rate, so placement and delivery are performed as described in the section mentioned above. At t=2.5, we can perform memory sharing between dedicated cache and multi-access cache schemes and achieve a rate lower than both of the schemes (dotted line in Figure 3, referring to scheme with memory sharing below the lines corresponding to dedicated cache scheme and scheme mentioned in Section 4.3).

However, it is important to acknowledge some limitations of our study. Firstly, we focused on noiseless broadcast links, which may not reflect real-world scenarios where channel impairments exist. Future research could investigate the impact of channel conditions on the performance of the proposed scheme. Additionally, we assumed non-colluding servers and replicated messages across the servers. Exploring the scheme’s resilience in the presence of adversarial behaviors or server failures could be an interesting direction for further investigation.

## 6. Conclusions

In this study, we introduced a PIR scheme that enables multiple users to securely retrieve data from distributed servers using a multi-access setup with cyclic wraparound cache access. We described the system model, formally defined the privacy and correctness constraints, and presented the transmission cost associated with our proposed scheme.

Our findings indicate that the multi-access setup with cyclic wraparound cache access offers significant advantages over the dedicated cache setup. By comparing the transmission costs of both setups, we demonstrated that the multi-access setup outperforms the dedicated cache setup, making it a more efficient and reliable approach for multi-user PIR scenarios. For instance, in Figure 2, for caching ratio M/N=0.25, we see that for L=2 the cyclic wraparound system and the dedicated cache system both achieve a rate of three. But the rate decreases as cache access degree *L* increases and users have access to more caches. For L=3,4,5,6 and 7, the rate of our scheme is 3,1.5,0.5,0.0625 and 0, respectively. More than a twofold improvement in download cost is shown compared to that of the dedicated cache setup from L=4 onward, i.e., accessing half of the available caches.

Furthermore, our scheme provides strong privacy guarantees, ensuring that users can retrieve data without revealing their individual retrieval patterns or compromising the privacy of the data. The proofs presented in Section 4.2.2 validate the privacy and transmission costs associated with our scheme, reinforcing its effectiveness and security. 

## Figures and Tables

**Figure 1 entropy-25-01228-f001:**
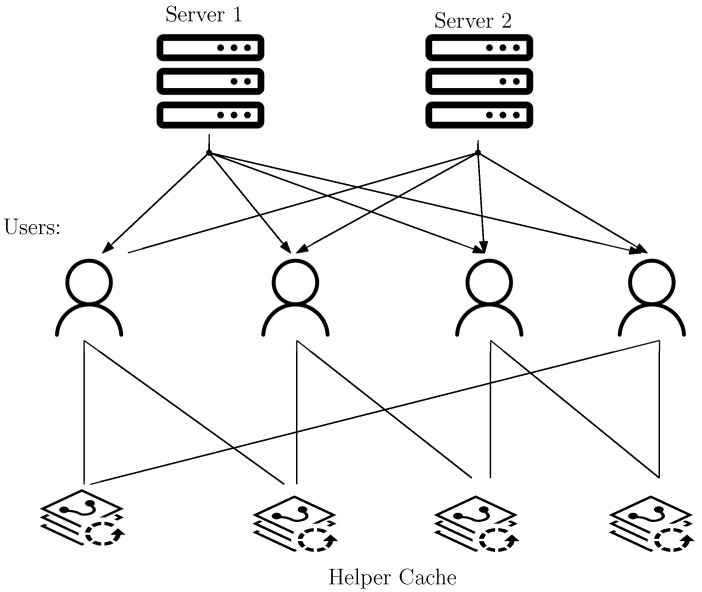
Multi-access coded caching setup with cyclic wraparound cache access with four users, four helper cache and two servers. Each user is accessing two adjacent helper caches.

**Figure 2 entropy-25-01228-f002:**
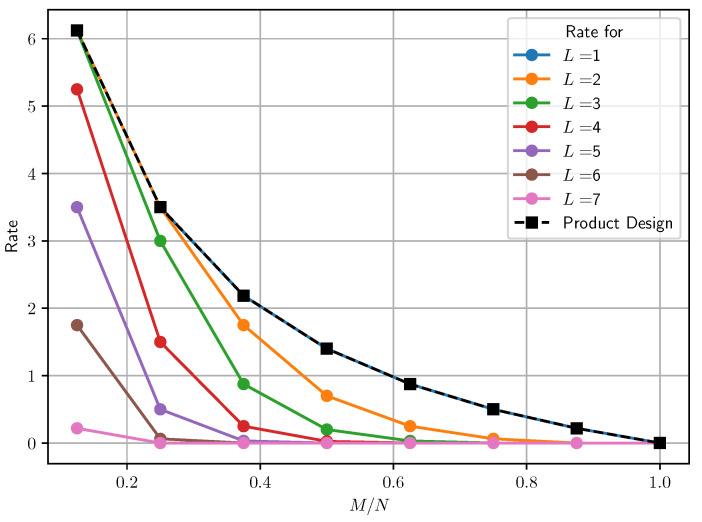
Comparison of transmission costs for dedicated cache (dotted lines) and multi-access (solid lines) with cyclic wraparound cache access. Here, we take K=8 users and cache nodes.

**Figure 3 entropy-25-01228-f003:**
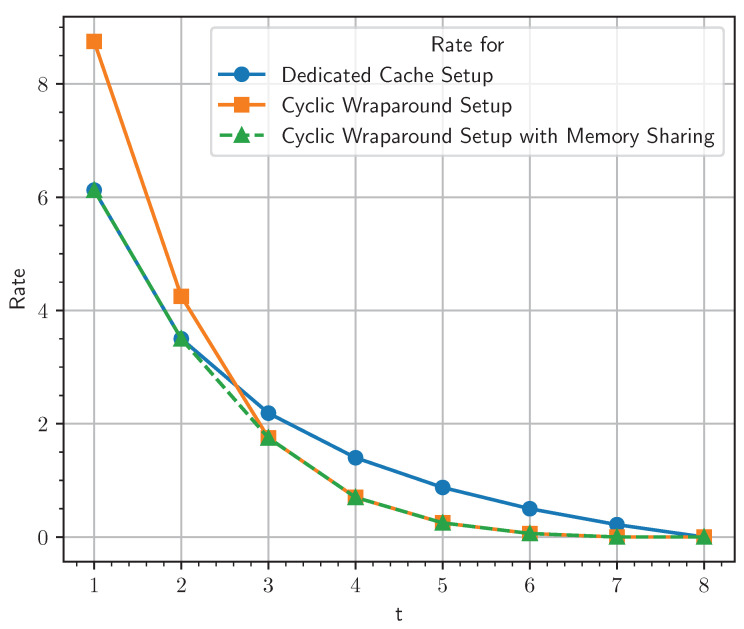
Transmission cost for K=8,L=2,S=2,N=3. Multi-access setup with cyclic wraparound cache access incur transmission cost only as high as dedicated cache setup with equal total memory in both systems.

## Data Availability

Data sharing not applicable.

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
