# Peer review of "Multi-User PIR with Cyclic Wraparound Multi-Access Caches [Author-notes fn1-entropy-25-01228]"

_entropy, 2023, doi:10.3390/e25081228_

Round 1
Reviewer 1 Report
The Introduction section must be improved. Block citations are not recommended. For these references, it must be detailed separately.
The paper includes a significant number of calculations, but it does not include any comparisons with the results obtained by other researchers who have conducted comparable research. In addition, there is neither a discussion nor any conclusions in the paper. This version of the work is ineligible for publication.
Author Response
Please see the pdf file uploaded.

Reviewer 2 Report
Figure 1 is the same as Figures 1 and 2 from the preprint https://arxiv.org/abs/2201.11481
Figure 2 matches the figure from the preprint https://arxiv.org/abs/2201.11481
In Figure 3, the text on the axes and in the legend should be enlarged. Figure 3 is the same as Figure 11 from the preprint https://arxiv.org/abs/2201.11481
The Acknowledgment introduces abbreviations that are not used further in the text.
The article lacks conclusions and directions for future research.
The title of Figure 1 should be capitalized.
Author Response
Please see the pdf file uploaded.

Reviewer 3 Report
Please make a thorough check for your abstract. It requires an improvement to highlight the paper contribution
The organization of the introduction confuses the reader; please refine the general flow of the paper
In section 1.5, the contribution is not well identified. Authors need to highlight their contribution in this paper
The complete story of the paper is confusing, how come the main results comes before the scheme
In section 3.1, what are the characterization of the dedicated cache considered by the authors?
The provided results are not discussed.
Authors provided one example as the scheme then brought the general one. Is this order logic. How could the example be contained in the general scheme, with which definitions, is it enough to take it as proof.
The paper is missing the conclusion
Figure 1 is not explained
Figure 3 is not clear, the axis and the text size are not readable
Equations sometimes numbered sometimes not
Before line 135, underline not needed
Sentences are not complete (e.g. Line 136, the sentence not complete, line 161, etc.)
Spelling and punctuation check is required.
The critical point is that the paper is not clear; the contribution is not well identified and highlighted. I am missing a thorough investigation of the state of the art. How could the authors confirm that their work outperforms available ones in the literature?
What is the improvement to the cited work in reference 22.
please check the punctuation and spelling.
Author Response
Please see the pdf file uploaded.

Round 2
Reviewer 1 Report
The paper was revised in accordance with the recommendation.
Reviewer 2 Report
Descriptions of all figures in the text are after the figures.
The seventh source is incorrectly specified:
A cache-aided PIR technique was recently put up by Ming et al. [7]
X. Zhang, K. Wan, H. Sun, M. Ji and G. Caire, “On the Fundamental Limits of Cache-Aided Multiuser Private Information Retrieval”, in IEEE Transactions on Communications, vol. 69, no. 9, pp. 5828-5842, Sept. 2021, doi: 10.1109/TCOMM.2021.3091612.
The abbreviation should not be used multiple times (for example, "Private Information Retrieval (PIR)"). Also, you should not enter abbreviations that are not used in the text below (for example, "Science and Engineering Research Board (SERB) of Department of Science and Technology (DST), Government of India, through J.C. Bose National Fellowship to B. Sundar Rajan, and by the Ministry of Human Resource Development (MHRD), Government of India, through Prime Minister's Research Fellowship (PMRF)").
The abbreviation "MAC-MuPIR" is only used in the annotation. There is a question to the abbreviation itself, why not "MACAMUPIR"?
Reviewer 3 Report
We thank the authors for considering the comments and suggestions from the preview. However, some points are still unclear and require thorough clarification, namely the following.
The discussion section is general. Instead, authors should discuss and quantify their results. Mainly how efficient their solution and in comparison to similar works.
Still find it confusing to bring the main results , than the scheme description. What is the logical relation between both. Are those results not suitable for the scheme provided in section 4.
What is the contribution from section 4 , why it is important to show these calculation, how it supports the work
For figure 2 and 3 in y-axis you put R, it is not clear to what it refers please be more specific.
As for English, it's fine. Authors should perform a thorough spelling and grammar check.
Round 3
Reviewer 3 Report
Some recommendation are in the follwoing:
better not to include reference in the conclusion. It is expected to have a summary of the most significant obtained results in the conclusion. For example support your indications in lines 464 and 465 with concrete values.
The titles of all sections still general, better to make them specific to your work.
in section 3, you are only having 1 subsection. it is useless if you don't have other subsection. Please improve it.
Please check the blance of your sections; Introduction over 4 pages ( too much), section 2 and 3 are soo small compared to section 4 which contains 8 subsections.
A check on the spelling and the grammar is necessary.
